# Vancomycin-Resistant *Enterococcus* Colonisation in the Patients of a Regional Spinal Cord Injury Unit in Northwest England, United Kingdom: Our Experience with Non-Isolation of VRE Colonised Patients

**DOI:** 10.3390/microorganisms13102257

**Published:** 2025-09-26

**Authors:** Vaidyanathan Subramanian, Bakulesh Madhusudan Soni, Gareth Derick Cummings, Sandra Croston, Kim Lucey, Ruth Hilton, Rachel Hincks

**Affiliations:** 1Northwest Regional Spinal Injuries Centre, District General Hospital, Southport PR8 6PN, UK; bakul.soni@merseywestlancs.nhs.uk (B.M.S.); sandra.croston@merseywestlancs.nhs.uk (S.C.); kim.lucey@merseywestlancs.nhs.uk (K.L.); ruth.hilton@merseywestlancs.nhs.uk (R.H.); 2Infection Prevention & Infection Control, District General Hospital, Southport PR8 6PN, UK; gareth.cummings@merseywestlancs.nhs.uk; 3Pharmacy, District General Hospital, Southport PR8 6PN, UK; rachel.hincks@merseywestlancs.nhs.uk

**Keywords:** spinal cord injury, Vancomycin-resistant Enterococcus, inpatients, healthcare, hand washing

## Abstract

We reviewed vancomycin-resistant *Enterococcus* (VRE) colonisation of inpatients of a spinal cord injury centre. The centre consists of one single occupancy en suite room and ten multi-occupancy rooms where two to six patients stay in a cubicle. These patients share bathroom and toilet facilities. Active screening for VRE is performed by taking rectal swabs on admission of patients to the spinal unit. The patients, who are colonised with VRE, are not isolated due to constraints in resources. During a twelve-month period (April 2024 to April 2025), 33 patients tested positive for VRE. In April 2025, 17 of 40 in-patients tested positive for VRE. During the last six 12-month periods from 2019, the number of patients testing positive for VRE has shown an upward trend from 18 during 2019–2020 to 33 during 2024–2025. No patient developed systemic infection with VRE (blood stream infection, endocarditis, meningitis, intra-abdominal sepsis, infection of a spinal implant or baclofen pump) during the study period. Twelve patients underwent implantation of a baclofen pump during 2024–2025. No patient developed VRE infection from the implant. We believe that non-isolation of patients colonised with VRE may be a pragmatic approach in a resource-poor healthcare facility. It is possible that non-isolation could have contributed to an increase in the number of patients who became colonised with VRE. Attention should be paid to infection prevention measures including hand washing and environmental cleaning to prevent the spread of VRE colonisation of inpatients and VRE infection of at-risk patients, e.g., immune-compromised individuals.

## 1. Introduction

Infections are prevalent after spinal cord injury (SCI) and constitute the main cause of death and are a rehabilitation confounder associated with impaired recovery. Patients with spinal cord injury can acquire a secondary, neurogenic immune deficiency syndrome characterised by reduced monocytic human leukocyte antigen-DR expression (mHLA-DR expression) and relative hypogammaglobulinaemia (combined cellular and humoral immune deficiency) [1].

### 1.1. Hospital-Acquired Infections in Patients with Spinal Cord Injury

Hospital-acquired or heath care-associated infections in are seen frequently persons with spinal cord injury [2]. The overall incidence of hospital-acquired infections in persons with spinal cord injury and disorder is higher than that reported for other populations, confirming the increased risk of hospital-acquired infections in persons with spinal cord injury [3].

A retrospective 4-year analysis of the postmortem data of 341 patients, who died after sustaining a spinal cord injury at a tertiary care apex trauma centre of India, revealed that out of 341 patients, the main cause of death in the patients with spinal cord injury was ascertained to be infection/septicemia, which occurred in 180 (52.7%) patients [4].

Enterococci can cause serious infections for patients in healthcare settings, including bloodstream, surgical site, and urinary tract infections. Nearly all vancomycin-resistant Enterococcus (VRE) infections happen in patients with healthcare exposures [5]. Persons with spinal cord injury are at risk of becoming colonised or infected with VRE because of prolonged hospital stay, presence of long-term indwelling urinary catheters, use of antibiotics for chest, urinary tract infections, and the many healthcare workers with whom they come in contact in a single day [6].

The primary transmission of vancomycin-resistant *Enterococcus* in the hospital setting is through the hands of healthcare providers. *Enterococcus* can persist on hands for as long as 60 min after inoculation and as long as four months on inanimate surfaces [7].

If vancomycin-resistant Enterococcus is picked up in a wound swab or from urine taken from a urinary catheter, or if a person is suffering with diarrhoea, the patient may be cared for in an isolation room [8]. In our Trust, the policy states, “You need to stay in a single room to prevent VRE being passed onto other patients who may be poorly or have little resistance to infection.” [9].

There is no provision for isolation of patients who test positive for vancomycin-resistant Enterococcus in the spinal injury unit due to constraints in resources. Only those patients who test positive for carbapenemase-producing Enterobacteriaceae, *Clostridioides difficile*, Nora virus, or COVID-19 novel coronavirus are isolated in the spinal unit.

### 1.2. Knowledge Gap

We reviewed colonisation of inpatients with VRE in our spinal unit to assess the impact of non-isolation of patients colonised by vancomycin-resistant Enterococcus. Such a study on the non-isolation of patients colonised with vancomycin-resistant Enterococcus in a spinal cord injury centre, to our knowledge, has not been published in PubMed. The findings will influence day-to-day operations, particularly in a resource-poor healthcare centre, where funding for creating isolation rooms is limited.

## 2. Materials and Methods

The Northwest Regional Spinal Cord Injury Unit is situated in the northwest of England, United Kingdom. This unit accommodates 43 inpatients with spinal cord injury. The unit contains eleven cubicles (rooms 1, 2, 4, 8, and 11 housing four patients each, rooms 3, 9, and 10 accommodating six patients each, rooms 5 and 7 housing two patients each and room 6 where only one patient stays). Patients with tracheostomy or those requiring ventilatory support are accommodated in rooms 1 and 3. Active screening of vancomycin-resistant Enterococcus was carried out by sending a rectal swab containing faecal matter for VRE from all patients on admission to the spinal cord injury unit.

The procedure was discussed with the patients and informed consent was taken for obtaining a rectal swab before a swab was obtained each time. No patient declined to give a swab for testing for vancomycin-resistant Enterococcus. After the initial swab taken at the time of admission, rectal swabs containing faecal matter for vancomycin-resistant Enterococcus were taken once a week from patients residing in rooms 1 and 3 and at monthly intervals for the remaining patients. However, some patients missed the active screening for VRE because they were not present in the ward when the swabs were taken (patient had gone on home leave or very rarely, swabs could not be taken on the due date because no faecal sample was present on the rectal swab).

VRE screening involves the culture testing of rectal swabs containing faecal material. Once the rectal swabs are cultured and the Enterococci species are identified, antibiotic resistance is determined for aminoglycoside, amoxicillin, co-trimoxazole, teicoplanin, vancomycin, tigecycline, and linezolid. This is then reported as R (Resistant), S (Susceptible) or I (Intermediate) per antibiotic.

## 3. Results

During a 12-month period of April 2024 to April 2025, 112 patients were admitted to the spinal unit. The data were obtained by analysing the information recorded in the National Spinal Cord Injury Database for the period of April 2024 to March 2025.

### 3.1. Demography of Spinal Unit Patients

The age profile and level of injury of these patients are depicted in Figure 1 and Figure 2.

The average length of stay of patients in the spinal unit during the year 2024–2025, grouped as per the level of spinal cord injury, is shown in Figure 3.

### 3.2. Antimicrobial Sensitivity of Enterococcus Species Isolated from Faecal Matter on the Rectal Swab

Table 1 shows the antimicrobial resistance totals of the first positive samples per organisms found. The data range is from April 2019 to March 2025. The definition of the abbreviations are R (Resistant), S (Susceptible), I (Intermediate), and N/A (No Data). N/A indicates samples where lab results did not show a result for that antibiotic. This data represents both rectal VRE screening and urine cultures positive for VRE, for both newly colonised patients on the unit and patients with a previous medical history of VRE colonisation from other care units, who have been transferred in.

### 3.3. Patients Testing Positive for Vancomycin-Resistant Enterococcus

A total of 33 patients (29%) tested positive for vancomycin-resistant Enterococcus on active screening. The sex distribution of these patients is shown in Figure 4A. Only a quarter of patients, who were colonised with VRE, were females.

The sex distribution of patients who were colonised with VRE as detected by rectal swab with faecal matter on active screening during the six-year period from April 2019 to March 2025 is given in Figure 4B. During the six-year period also, the patients who were colonised with VRE as detected by a rectal swab containing faecal matter was 23%.

### 3.4. Symptomatic Infection with Vancomycin-Resistant Enterococcus

No patient developed a symptomatic infection with vancomycin-resistant Enterococcus.

Nine out of thirty-three patients tested positive for vancomycin-resistant Enterococcus within three days of their admission to the spinal unit. These patients could have become colonised with vancomycin-resistant Enterococcus during their stay in the previous healthcare facility and might have been positive for vancomycin-resistant Enterococcus even prior to their admission to the spinal unit.

At present (April 2025), 17 patients in the spinal unit tested positive for vancomycin-resistant Enterococcus in their rectal swab. One of these patients showed growth of vancomycin-resistant Enterococcus in their urine as well. None of these patients showed features of systemic infection by VRE, e.g., endocarditis, meningitis, intra-abdominal abscess. Six patients have diabetes; five are taking insulin; and the sixth patient is taking metformin.

### 3.5. Case Vignettes

A 74-year-old male person with tetraplegia, atrial fibrillation, hypertension, pericarditis, liver cirrhosis, epidural abscess (drained), proctitis, tracheostomy, and chronic obstructive pulmonary disease, tested positive for vancomycin-resistant Enterococcus on 23 April 2018 and continues to be colonised with VRE on 10 July 2025. This patient has been staying in the spinal unit as suitable accommodation and care requirements after discharge have not been sorted out yet. VRE was resistant to teicoplanin. When the C-reactive protein increased from 12 mg/L on 3 March 2025 to 79 mg/L on 7 March 2025, a microbiologist advised tigecycline. This patient received 100 mg on 8 March 2025 followed by three doses of 50 mg every 12 h until 9 March 2025. This patient continues to be colonised with vancomycin-resistant Enterococcus. This case illustrates that colonisation by vancomycin-resistant Enterococcus may persist in some spinal cord injury patients for months or even years. Detection of vancomycin-resistant Enterococcus in urine is often a red herring in spinal cord injury patients and the physician should look for a source of sepsis elsewhere in the body, e.g., chest infection, infected pressure sore, etc.

We illustrate this by what we observed with another patient. A 57-year-old male patient with tetraplegia, diabetes mellitus, hypertension, ischaemic heart disease, cardiac stent in place, and bronchial asthma, developed a temperature of 38.5 degrees Celsius very briefly. C-reactive protein in blood was only 4 mg/L (within the reference range). Blood culture showed no growth. On an empirical basis, this patient was prescribed meropenem, but the antibiotic was changed to tigecycline because the urine culture had shown growth of *Enterococcus faecium* resistant to vancomycin. Both rectal swab and urine specimen continued to show growth of vancomycin-resistant Enterococcus while the patient was receiving tigecycline and afterwards.

This patient received tigecycline 100 mg stat followed by 50 mg twice a day (nine doses) for a total of five days. Urine showed growth of VRE twelve days, four days, and two days before commencing tigecycline. Urine continued to show the presence of VRE two days after starting tigecycline, four days, eleven days, eighteen days, twenty-six days, thirty-three days, thirty-four days, thirty-nine days, forty-six days, fifty-three days, and sixty-seven days after completing the course of tigecycline. Interestingly, the rectal swab of this patient was negative for vancomycin-resistant Enterococcus four days before the patient received tigecycline. But two days after commencing tigecycline, the rectal swab showed growth of VRE.

In hindsight, we realised that the presence of vancomycin-resistant Enterococcus in urine was probably a red herring. The microbiology report of the urine culture stated clearly, “Growth of VRE may reflect colonisation of catheter. Suggest do NOT treat with antimicrobials unless clinical infection, e.g., sepsis, fever, rigours, new confusion, flank/suprapubic pain or tenderness.”

Tigecycline, a glycylcycline antimicrobial derived from minocycline, exerts activity against vancomycin-resistant Enterococcus by inhibiting the 30S ribosomal subunit, thereby blocking protein synthesis [10]. Microbiological cure rate may not be achieved due to many factors, for example, vancomycin-resistant Enterococcus can form biofilms in the urinary tract, which can protect the bacteria from antibiotics and hinder eradication. In persons with spinal cord injury and long-term indwelling urinary catheters, the presence of vancomycin-resistant Enterococcus is often a red herring. When a patient develops a temperature, the physician should look for a source of sepsis in the whole body and should not be carried away by a positive urine culture report of vancomycin-resistant Enterococcus as patients are often colonised with VRE. Even after administration of antibiotics, the cure rate may not be dissimilar to the non-treated patients. In an 18-month evaluation of patients with vancomycin-resistant Enterococcus isolated from urine cultures at a tertiary care hospital, microbiologic cure rates among symptomatic patients who received antibiotics (nitrofurantoin, ciprofloxacin, or chloramphenicol [9 patients]) and those who did not (four patients) were similar (67% vs. 50%) [11].

### 3.6. Baclofen Pump Implants and Risk of Infection with VRE

During the study period, twelve patients underwent implantation of a new baclofen pump or removal of the old pump and implantation of a replacement pump. Eleven patients did well and had no problem. A 64-year-old female patient developed a wound infection following the replacement of a baclofen pump. As a precautionary measure, the pump was removed. This patient tested negative for vancomycin-resistant Enterococcus during active screening. The wound swab showed no growth.

### 3.7. Number of Patients Colonised with VRE in the Spinal Unit over Six 12-Month Periods

The number of patients who were newly colonised by vancomycin-resistant Enterococcus over the last six 12-month periods from 2019 to 2020 are shown in Figure 5A,B.

During 2019–2020, 18 inpatients were colonised with vancomycin-resistant Enterococcus. But in 2024–2025, 33 inpatients tested positive for vancomycin-resistant Enterococcus.

The number of patients newly colonised with vancomycin-resistant Enterococcus was 23 during 2021–2022 but increased to 41 during 2022–2023 (Figure 5A). There was a substantial increase in the number of patients aged 60 years and above admitted to the spinal unit during 2022–2023 (87 patients) as compared to 2021–2022 when only 53 patients were admitted (Figure 5B). The number of patients newly colonised with vancomycin-resistant Enterococcus, when expressed as a percentage of the number of patients aged 60 years and above, remained 43 and 47, respectively. Thus, the increase in the number of patients newly colonised with vancomycin-resistant Enterococcus during 2022–2023 is probably due to an increase in the number of patients above the age of 60 years admitted to the spinal unit. The older patients tend to have an increased number of long-term conditions, e.g., diabetes, and manage the urinary bladder by indwelling urinary catheters.

### 3.8. Duration of Colonisation with Vancomycin-Resistant Enterococcus

The duration of colonisation with vancomycin-resistant enterococcus varied widely. Ninety-one patients were colonised for one month whereas in nine patients, vancomycin-resistant enterococcus were positive for more than twelve months. The average length of colonisation of vancomycin-resistant enterococcus for patients over the period from April 2019 to March 2025 was 3.568 months—so rounded up, it would be 4 months (Figure 6).

### 3.9. Spinal Unit Patients Testing Positive for VRE in Urine

The number of patients newly colonised with VRE in urine also showed an increasing trend between 2019–2020 and 2024–2025. The number of patients who were newly colonised by vancomycin-resistant Enterococcus in urine over the last six 12-month periods from 2019 to 2020 are shown in Figure 7. Only six patients were newly colonised with vancomycin-resistant Enterococcus in urine during 2019–2020. But the number of patients who were newly colonised with vancomycin-resistant Enterococcus in urine during 2024–2025 increased 2.5-fold to 15.

The sex distribution of these patients who were newly colonised by vancomycin-resistant Enterococcus in urine is shown Figure 8. Three-fourths of the patients who were newly colonised with vancomycin-resistant Enterococcus are males. Female patients account for only 26%.

### 3.10. Annual Expenditure on Antibiotics Prescribed to Patients in the Spinal Unit During the Six 12-Month Periods

During the same period, the annual expenditure on antibiotics prescribed to patients in the spinal unit almost doubled (Figure 9) from £11,919 to £20,474. The use of piperacillin/tazobactam increased from 678 vials during 2019–2020 to 855 during 2024–2025 (Figure 10). In contrast, the prescription for meropenem decreased from 1861 vials during 2019–2020 to 1320 vials during 2024–2025 (Figure 11). The number of 1 g vials of vancomycin for intravenous use issued to spinal unit decreased from 122 during 2019–2020 to 20 during 2024–2025 (Figure 12).

Summarising, the prescription for vancomycin (intravenous) and meropenem decreased between the years 2019–2020 and 2024–2025. Despite the more judicious use of vancomycin and meropenem in the spinal ward, the number of patients, who became colonised with vancomycin-resistant Enterococcus, increased from 2019–2020 to 2024–2025.

## 4. Discussion

Dumford and associates [12] studied 22 hospitalised male patients with spinal cord injury in Cleveland, OH, USA. Of the 22 asymptomatic patients, 12 (55%) were carriers of vancomycin-resistant Enterococcus in rectal cultures obtained by introducing a sterile swab into the rectum. These authors concluded that asymptomatic carriage of vancomycin-resistant Enterococcus was common among patients with spinal cord injury. Cultures of skin and environmental surfaces were collected concurrently with stool cultures from asymptomatic faecal VRE carriers. A total of 25% of asymptomatic carriers of vancomycin-resistant Enterococcus in rectal cultures tested positive in the skin samples. Two of the twelve patients’ environment (one each of bed rail and hand grip) tested positive for vancomycin-resistant Enterococcus.

In our spinal unit (where patients colonised with vancomycin-resistant Enterococcus are not isolated), currently (April 2025), 17 patients (42.5%) are colonised with vancomycin-resistant Enterococcus in contrast to the 55% patients becoming colonised with vancomycin-resistant Enterococcus in Cleveland as reported by Dumford [12].

Among seventeen patients who are presently in-patients of the spinal unit and positive for vancomycin-resistant Enterococcus, six have diabetes mellitus (35%) and five of these patients have been taking insulin.

Persons with spinal cord injury are at risk for becoming colonised with vancomycin-resistant Enterococcus because of the following risk factors: (1) indwelling urinary catheters; (2) previous antimicrobial therapy (for chest or urinary tract infection) and sometimes, extended use of antibiotics, say for spondylo-discitis; (3) long hospital stay; (4) proximity to other patients with vancomycin-resistant *Enterococcus*.

No patient in the spinal unit developed symptomatic infection by vancomycin-resistant Enterococcus, e.g., blood stream infection, central nervous system infection or endocarditis. However, non-isolation of patients colonised with vancomycin-resistant Enterococcus could have been a predisposing factor for the spread of VRE in patients in the same cubicle. We observed that the patients colonised with vancomycin-resistant Enterococcus are often seen in clusters. Two out of four patients in room 1 are colonised with vancomycin-resistant Enterococcus; four patients out of six patients in room 3 are colonised with VRE; three patients out of four patients in room 8 are colonised; two of six patients staying in room 9 are colonised with VRE. If patients who are colonised with vancomycin-resistant Enterococcus are isolated, it is possible that the transmission of VRE among patients within a cubicle might have been curtailed to some extent.

### 4.1. Possible Role of Bowel Evacuation on the Bed in the Spread of VRE Among Other Patients Staying in the Same Cubicle

We believe that bowel evacuation on the bed may be a risk factor for the spread of vancomycin-resistant Enterococcus among persons with spinal cord injury. Some patients in the spinal unit require bowel care in the bed because they need to stay in bed, e.g., due to a pressure sore. Bowel evacuation on the bed may be a unique risk factor for the spread of vancomycin-resistant Enterococcus among patients in a spinal unit. Such a situation will be encountered very rarely in a ward housing persons with medical illness. Preferably, enhanced ventilation should be provided in these rooms where bowel evacuation is performed on the bed, almost like the standards of ventilation recommended for special areas in the hospital premises including toilets to diminish the risk of bioaerosol concentrations. [13] It is not possible to keep the patients who require bowel evacuation in one or two dedicated rooms. Patients are housed together based on their sex, and clinical conditions, viz., whether a patient has a tracheostomy or requires ventilatory support. Ideally, all patients should stay in single occupancy en suite rooms. Provision of single occupancy rooms for all patients will provide privacy and reduce the risk of cross infection with hospital-acquired pathogens. But many hospitals built during the last century do not contain such facilities.

### 4.2. Duration of VRE Colonisation in Spinal Unit Patients

The average length of colonisation of vancomycin-resistant enterococcus for patients over the period from April 2019 to March 2025 was 3.568 months—so rounded up, it would be 4 months. We did not study the duration of colonisation by vancomycin-resistant Enterococcus after discharge from the spinal unit. Sohn and associates [14] investigated the duration of colonisation with vancomycin-resistant Enterococcus and the risk factors for prolonged carriage in the outpatient clinic after discharge from the hospital. Median duration of vancomycin-resistant Enterococcus colonisation was 5.57 weeks after hospital discharge. A retrospective cohort study to understand the duration of carriage of vancomycin-resistant Enterococcus (by screening of a single stool culture) and associated factors among patients who had been identified with vancomycin-resistant Enterococcus infection and/or colonisation since the year 2000 in Australia showed that although the duration of carriage of detectable colonisation can be as long as 4 years, a significant proportion of discharged patients either clear or cease shedding VRE in their faeces after 1 year [15].

A male patient with tetraplegia in our spinal unit continues to be colonised with vancomycin-resistant Enterococcus for seven years since 2018. Baden and associates [16] observed persistence of colonisation with vancomycin-resistant *Enterococcus faecium* in the nononcologic, non-intensive care unit patient. Colonisation with vancomycin-resistant Enterococcus may persist for years, even if the results of intercurrent surveillance stool and index site cultures are negative. 

### 4.3. Number of Patients Who Are Colonised with VRE and the Number of Patients Above 60 Years of Age in the Spinal Unit—Is There a Positive Relationship?

From 2019 to 2020, the number of inpatients of the spinal unit who test positive for vancomycin-resistant Enterococcus during every 12-month period fluctuated from 18 to 41 patients. But the overall trend has been an upward trajectory. It is interesting to note that there was a gradual increase in the number of older patients with multiple long-term conditions getting admitted to the spinal unit during this period (Figure 13).

### 4.4. Increasing Number of Patients with VRE Colonisation and the Use of Antibiotics in the Spinal Unit

We observed an increase in the number of patients who became colonised with Vancomycin-resistant Enterococci between 2019 and 2025. An increase in the number of isolates of vancomycin-resistant Enterococcus has been observed both in the United States of America and in Europe. Fitzpatrick and associates [17] studied 216,504 isolates from 19,421 patients (adults with spinal cord injury/disorder treated at VA Medical Centres (VAMCs) between 1 January 2005 and 31 December 2013). The percentage of *E. faecalis* and *E. faecium* isolates that were vancomycin-resistant increased over time with greater odds of vancomycin resistance in *E. faecalis* and *E. faecium* in 2013 compared with 2005. These researchers stated that the possible explanations might be increased antibiotic exposures and increased healthcare exposures.

The European Antimicrobial Resistance Surveillance Network (EARS-Net) by 30 European Union (EU) and European Economic Area (EEA) countries in 2019 (data referring to 2018), and trend analyses of data reported by the participating countries for the period of 2015 to 2018 [18] showed an increase in the EU/EEA population-weighted mean percentage for vancomycin-resistant *Enterococcus faecium* from 10.5% in 2015 to 17.3% in 2018. Contrary to many other species under surveillance, no distinct geographical pattern could be seen for vancomycin-resistant *E. faecium*, as high percentages were reported from Southern, Eastern and Northern Europe. 

During the same period (from 2019–2020 to 2024–2025), we observed an increase in the number of patients colonised with vancomycin-resistant Enterococcus; there was an increase in the amount of money spent on antibiotics prescribed to patients in the spinal unit from £11,919 to £20,474 (Figure 9). We used the cost of antibiotics as a surrogate marker for the use of antibiotics in the patients of the spinal unit.

Correa-Martinez and associates [19] studied the risk factors for the persistence of vancomycin-resistant Enterococcus between 2016 and 2018 in the 1527-bed University Hospital Münster, which is a tertiary care centre, admitting 65,000 patients every year. Use of antibiotics was a risk factor for the persistence of vancomycin-resistant Enterococcus. Use of penicillin, ampicillin, amoxicillin, ampicillin/sulbactam, amoxicillin/clavulanic acid, flucloxacillin, cefuroxime, erythromycin, rifampicin and fosfomycin was not a risk factor. But piperacillin/tazobactam, ceftriaxone, ciprofloxacin, clindamycin, trimethoprim/sulfamethoxazole and understandably, vancomycin use were risk factors for the persistence of vancomycin-resistant Enterococcus. Figure 10 and Figure 11 show the use of piperacillin/tazobactam and meropenem in the spinal unit during six consecutive twelve-month periods.

Most importantly, the duration of treatment with antibiotics influenced the persistence of vancomycin-resistant Enterococcus. Patients in whom vancomycin-resistant Enterococcus persisted received antibiotics for 97.7 days. But the patients, in whom clearance of vancomycin-resistant Enterococcus was observed, received antibiotics only for 25 days [18].

In the spinal unit, we have been practicing antimicrobial stewardship as per the UK Health Security Agency Guidance “Start smart then focus: antimicrobial stewardship toolkit for inpatient care settings” [20]. The prescription for meropenem and intravenous vancomycin decreased between the years 2019–2020 and 2024–2025. Despite the judicious use of intravenous vancomycin and meropenem in the spinal unit, the number of patients who were colonised with vancomycin-resistant Enterococcus increased from 18 during 2019–2020 to 33 2024–2025. This shows the multifactorial aetiology for the occurrence of vancomycin-resistant Enterococcus in patients. It has been shown that urinary catheters play a vital role in the prevalence of vancomycin-resistant Enterococcus in healthcare settings. Biofilm formation over the urinary catheter acts as a major contributing factor in establishment of Enterococcal infection [21].

Šámal and associates [22] studied the prevalence of antibiotic-resistant and multidrug-resistant bacteria in urine cultures from inpatients with spinal cord injuries and disorders for an 8-year period. The patients were hospitalised with spinal cord injury/disorder in the spinal care ward from 1 January 2013 to 31 December 2020. Data was obtained from electronic medical records (EMRs) and the central database of the microbiology department. A total of 396 patients were evaluated (303 men and 93 women) and 1101 urine samples were collected from which 1385 bacterial isolates were obtained. These researchers did not observe the methicillin-resistant *Staphylococcus aureus* or vancomycin-resistant Enterococcus strains. The researchers did not provide reasons for their findings.

### 4.5. Increasing Number of Patients Newly Colonised with VRE in Urine in the Spinal Unit from 2019 to 2025

We observed an increase in the number of patients newly colonised with vancomycin-resistant Enterococcus in urine between 2019–2020 and 2024–2025. Only six patients were newly colonised with vancomycin-resistant Enterococcus in urine during 2019–2020. But the number of patients who were newly colonised with vancomycin-resistant Enterococcus in urine during 2024–2025 increased 2.5-fold to 15 (Figure 7). The sex distribution of patients who were colonised in the urine was like the sex distribution of patients in whom the rectal swab was positive for VRE. We expected an increased percentage of female patients to become colonised in the urine because of their anatomy. Probably, this reflected the good perineal hygiene and care of indwelling urethral catheters delivered to these patients in the spinal unit.

### 4.6. Isolation of Patients with VRE Colonisation in the Spinal Unit—A Pragmatic Approach

Isolation of spinal cord injury patients, who are in the process of undergoing rehabilitation, may curtail their level of participation in the gym, occupational therapy, and physiotherapy departments. This may in turn lead to an increase in the length of stay of these patients in the spinal unit as these patients may take a longer period to achieve the rehabilitation goals set for them.

Prang and associates [23] found that prolonged isolation due to multi-drug-resistant organism colonisation for over a third of the inpatient comprehensive care period did not appear to impair neurological recovery and functional outcome within the first year after spinal cord injury. Isolation-related measures mainly restricted the use of a variety of machines required for interventions at the body structure/function (e.g., machine-based endurance and muscular strength training) and activity level (e.g., body weight supported treadmill or practicing the transfer from the wheelchair into a car). The patients who were isolated continued to receive exercises such as using resistance bands or weights, arm/leg cycling training which were brought into the patient room, and basic activity level (e.g., parallel bars for walking training or, arm suspension devices for hand/arm use in a room reserved for isolated patients). These patients missed training related to household tasks in the patient-adapted kitchen. Freed-up therapy slots were filled with rehabilitative interventions, which could be applied despite isolation.

Additional resources in terms of space, staffing, and equipment are required to provide such a level of rehabilitation to patients who are isolated in a spinal unit. Some spinal units including our unit will not be able to provide point-of-care rehabilitation due to constraints in physical and human resources.

### 4.7. Treatment of VRE Colonisation

We did not treat patients who were colonised with vancomycin-resistant Enterococcus with antibiotics. Kriz and associates [24] carried out faecal microbiota transplantation in two patients of the Spinal Cord Unit at University Hospital Motol, who were carriers of vancomycin-resistant Enterococci. Donor faeces were obtained from healthy, young, screened volunteers. A total of 200–300 mL of suspension was applied through a nasoduodenal tube. Decolonisation was achieved 27 and 55 days after faecal microbiota transplantation. We observed spontaneous clearance of vancomycin-resistant Enterococcus in some patients in our spinal unit. Therefore, it is difficult to say with certainty that faecal microbiota transplantation was the cause for decolonisation in the two patients. The National Institute for Health and Care Excellence in United Kingdom has not recommended faecal transplantation as a possible method of achieving decolonization in persons with spinal cord injury, who are colonised with vancomycin-resistant Enterococcus.

### 4.8. Interhospital Transfer of Spinal Cord Injury Patients with VRE Colonisation

In the present study, nine of thirty-three patients tested positive for vancomycin-resistant Enterococcus within three days of their admission to the spinal unit. These patients could have become colonised with vancomycin-resistant Enterococcus during their stay in the previous healthcare facility and may represent interhospital transmission of vancomycin-resistant Enterococcus. Saito and associates [25] reported interhospital transmission of vancomycin-resistant Enterococcus faecium in six general hospitals in the Aomori prefecture, Japan. These researchers suggested stopping transfers of vancomycin-resistant Enterococcus patients and contact patients to other units or to any other hospitals. But it may not be feasible in case of patients with spinal cord injury who need to come to a specialist rehabilitation unit to achieve the best outcome after a devastating spinal cord injury. However, we diligently adhered to national evidence-based guidelines for preventing healthcare-associated infections (HCAI) in National Health Service (NHS) hospitals in England [26].

### 4.9. Summary of the Real-World Experience from a Spinal Injuries Unit on VRE Colonisation

The number of inpatients in the spinal unit, who become colonised with vancomycin-resistant Enterococcus showed an upward trend from 2019 to 2020. Eighteen patients were colonised with vancomycin-resistant Enterococcus in 2019–2020 whereas 33 patients tested positive for vancomycin-resistant Enterococcus during 2024–2025.In April 2025, 17 patients (42.5%) in the spinal unit tested positive for vancomycin-resistant Enterococcus. No patient developed systemic infection (blood stream infection or central nervous system infection or endocarditis) due to vancomycin-resistant Enterococcus.The number of patients newly colonised with vancomycin-resistant Enterococcus in urine also showed a 2.5-fold increase between 2019–2020 and 2024–2025.Due to constraints in resources, patients colonised with vancomycin-resistant Enterococcus are not isolated in our spinal unit. But no patient came to harm because of non-isolation.One patient who tested positive for vancomycin-resistant Enterococcus in 2018 has remained positive for VRE continuously for seven years. The last rectal swab taken in April 2025 was positive for vancomycin-resistant Enterococcus. This case illustrates the challenges in eradicating vancomycin-resistant Enterococcus in patients with spinal cord injury. Even treatment with linezolid is unlikely to eradicate vancomycin-resistant Enterococcus. This is reflected in our Trust policy which states, “Those who are infected can be treated with an antibiotic called Linezolid. This does not eradicate the organism completely; it just cures the infection. The organism will continue to colonise. Therefore, precautions will still need to be taken to prevent spread of infection, even after the treatment course has been completed.” [9].Twelve patients underwent implantation of a baclofen pump during 2024–2025. No patient developed vancomycin-resistant Enterococcus infection of the implant while we followed non-isolation of patients colonised with vancomycin-resistant Enterococcus.

## 5. Conclusions and Take-Home Messages

About 40% of patients in the spinal unit tends to be colonised with vancomycin-resistant Enterococcus. This may reflect upon the current architecture of spinal unit where most of the patients are accommodated in multi-occupancy patient cubicles and share toilet facilities. Single occupancy en suite rooms may help to curtail transmission of vancomycin-resistant Enterococcus to some extent.

We did not isolate patients who are colonised with vancomycin-resistant Enterococcus in the spinal unit. A pragmatic approach may be the non-isolation strategy with emphasis on infection prevention measures. The document “Control and Management of patients Colonised or Infected with Vancomycin resistant enterococci (VRE) Guideline” is a useful aid [27].

All health professionals should diligently follow “Your 5 moments for hand hygiene at the point of care” as recommended by the National Health Service, United Kingdom [28] and World Health Organisation [29].

Treatment of vancomycin-resistant Enterococcus colonisation in spinal cord injury patients, who are not immune compromised is not required. Presence of vancomycin-resistant Enterococcus in the urine of a patient who develops a temperature may not necessarily indicate urinary tract infection in the absence of other collaborating symptoms and findings.

### Epilogue

Human vancomycin-resistant Enterococcus prevalence was associated with vancomycin-resistant Enterococcus in both livestock and in food products (*p* < 0.001, *p* = 0.04, respectively). Meta-regression demonstrated that human vancomycin-resistant Enterococcus prevalence increased at a rate of 0.75% (95% CI: 0.46–1.04%; *p* < 0.001) per 1% increase in livestock VRE colonisation [30]. The meta-analysis established a clear link of vancomycin-resistant Enterococcus across One Health sectors.

A study of the occurrence, antimicrobial resistance profiles, virulence factors, and plasmid composition of the *Enterococcus* species isolated from salad ingredients in the United Arab Emirates revealed an *Enterococcus* detection rate of 85.5% (342/400). Among 85 *Enterococcus* isolates tested for antimicrobial susceptibility, 55.3% displayed resistance to at least one agent, with 18.8% classified as multidrug-resistant [31].

Vancomycin-resistant Enterococcus colonisation is likely to be elevated for those in contact with colonised animals or contaminated food products. Although primarily hospital-associated, vancomycin-resistant *Enterococcus faecium* are also detected in the community, food chain, livestock, and environmental sources like wastewater, indicating diverse transmission pathways [32]. Therefore, there is a need for the One Health approach in eradication of vancomycin-resistant Enterococcus in patients staying in healthcare facilities.

## Figures and Tables

**Figure 1 microorganisms-13-02257-f001:**
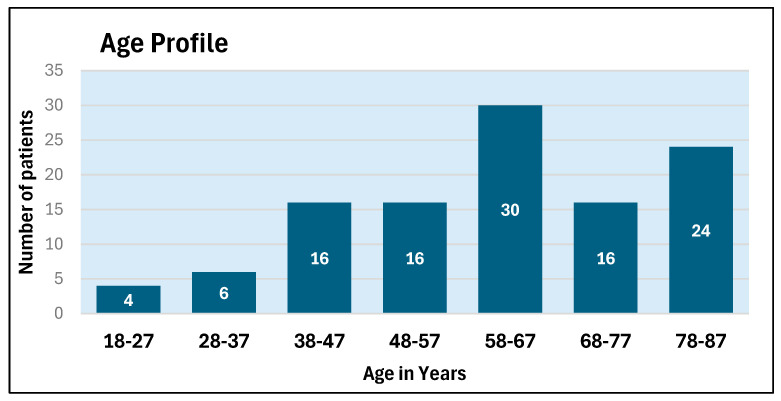
Age profile of patients admitted to the spinal unit during 2024–2025. (X axis: Age in years).

**Figure 2 microorganisms-13-02257-f002:**
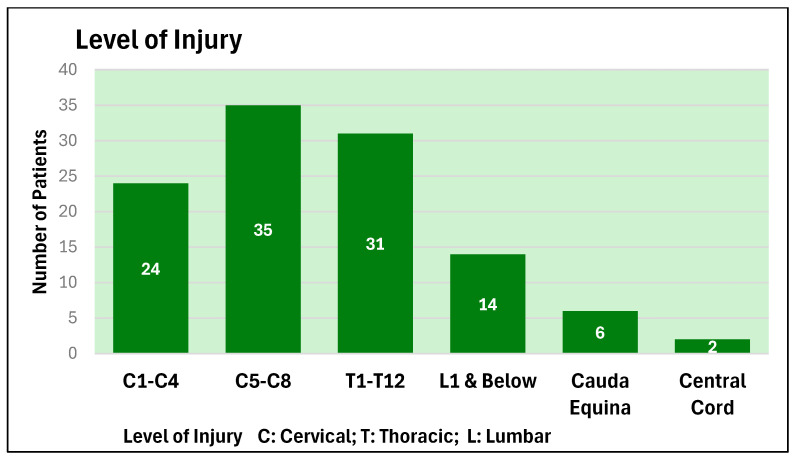
Level of injury as per spinal segments (C: Cervical; T: Thoracic; L: Lumbar) of the patients who were admitted to the spinal unit during 2024–2025.

**Figure 3 microorganisms-13-02257-f003:**
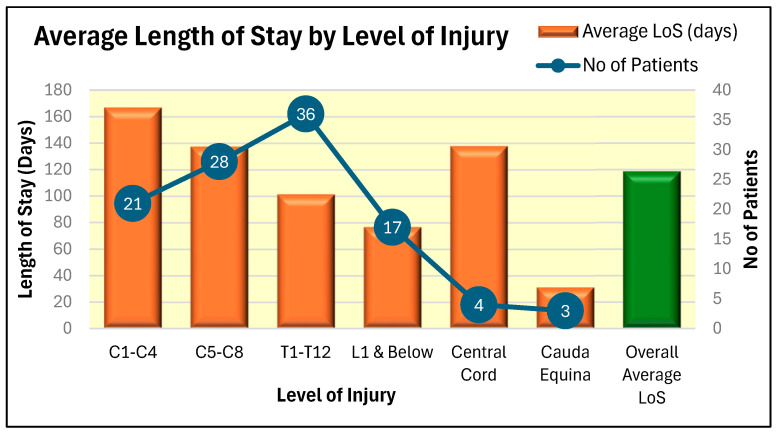
Average Length of Stay (LoS) of patients in the spinal unit according to the level of injury for years 2024–2025. (C: Cervical; T: Thoracic; L: Lumbar).

**Figure 4 microorganisms-13-02257-f004:**
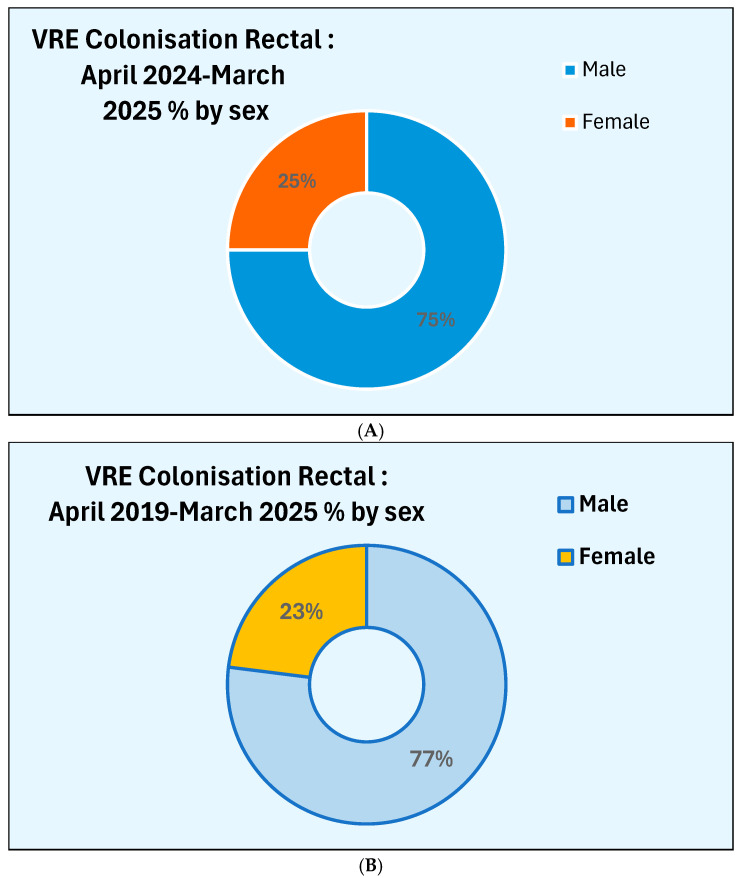
(**A**): Sex distribution of patients newly colonised from April 2024 to March 2025. (**B**): Sex distribution of patients newly colonised from April 2019 to March 2025.

**Figure 5 microorganisms-13-02257-f005:**
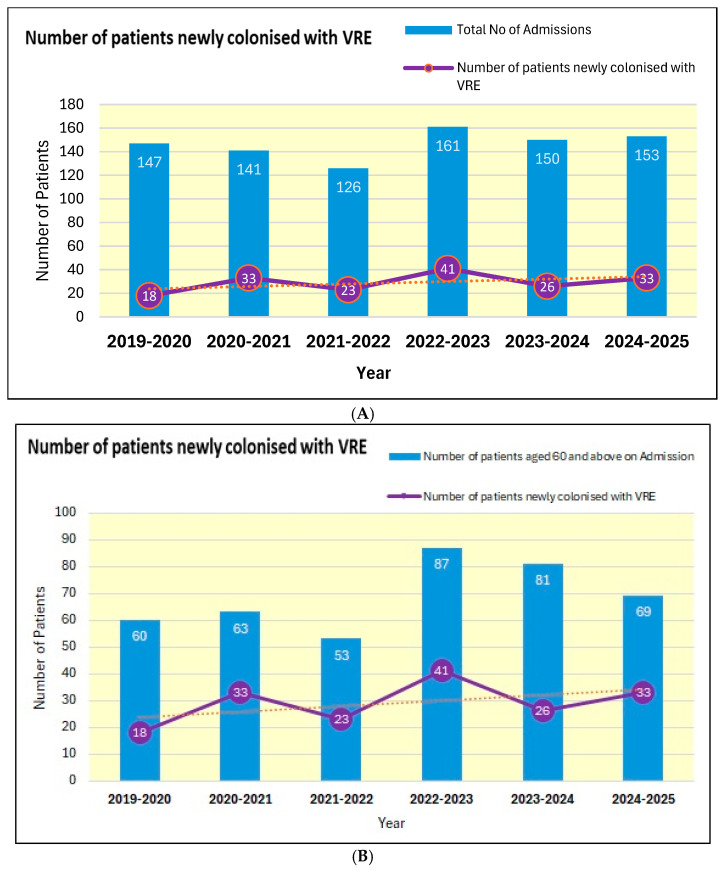
(**A**): Number of patients newly colonised with vancomycin-resistant Enterococcus for six 12-month periods from 2019 to 2025 versus the total number of admissions during the same periods. (X axis: Years in twelve-month period of April to March). (**B**): Number of patients newly colonised with vancomycin-resistant Enterococcus for six 12-month periods from 2019 to 2025 versus the number of patients aged 60 and above on admission.

**Figure 6 microorganisms-13-02257-f006:**
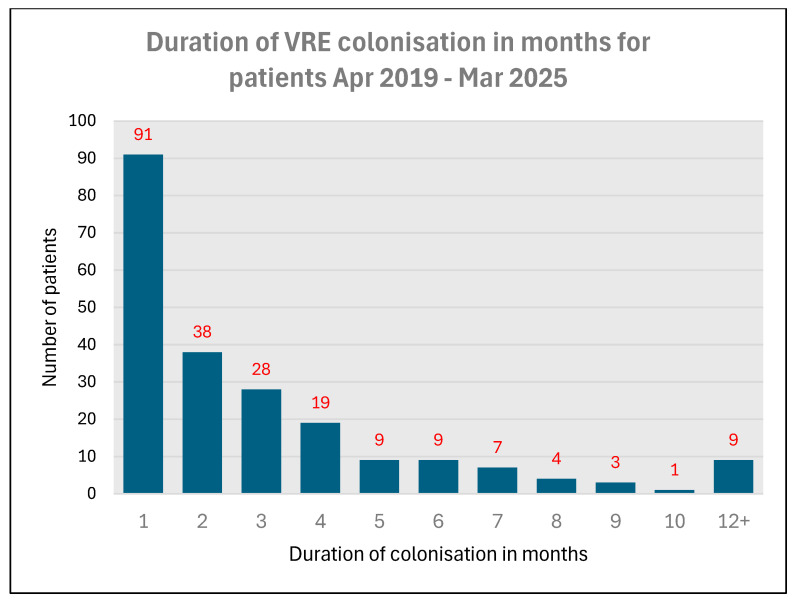
Duration of colonisation with vancomycin-resistant Enterococcus in spinal unit patients during 2019–2025.

**Figure 7 microorganisms-13-02257-f007:**
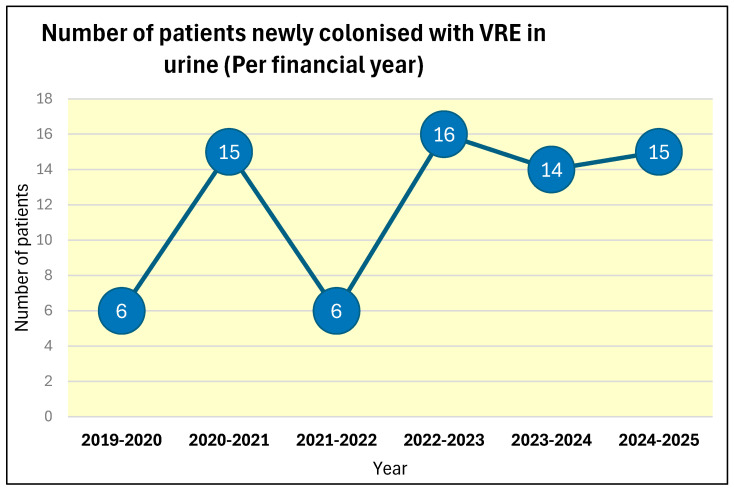
Number of patients newly colonised with vancomycin-resistant Enterococcus in urine during the six twelve-month period from 2019–2020 to 2024–2025.

**Figure 8 microorganisms-13-02257-f008:**
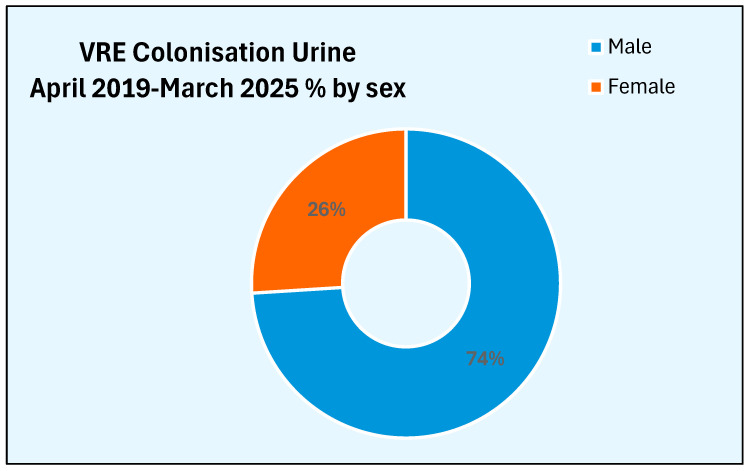
Sex distribution of patients who were newly colonised with VRE in urine.

**Figure 9 microorganisms-13-02257-f009:**
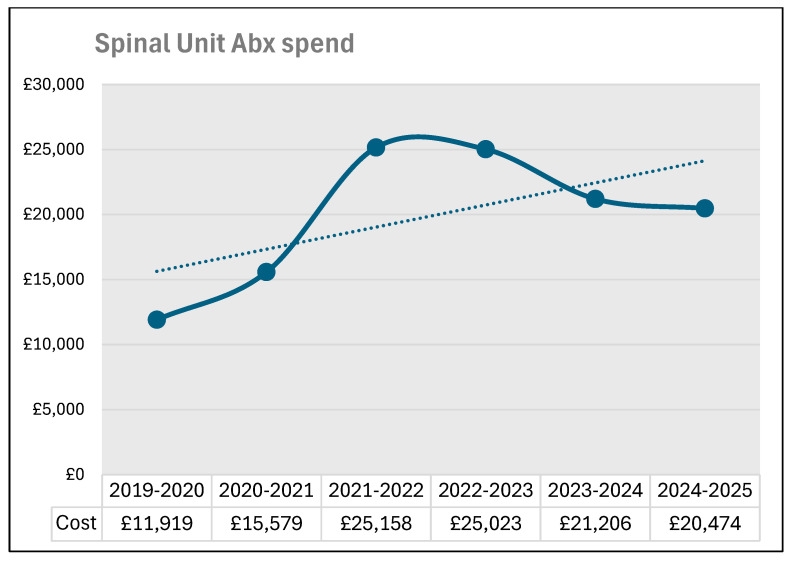
Annual expenditure on antibiotics in the spinal unit from 2019–2020 to 2024–2025.

**Figure 10 microorganisms-13-02257-f010:**
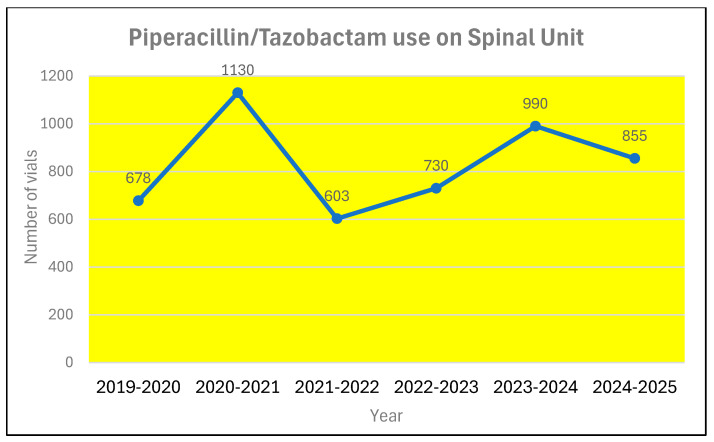
Number of vials of piperacillin/tazobactam issue to spinal unit during six consecutive twelve-month periods.

**Figure 11 microorganisms-13-02257-f011:**
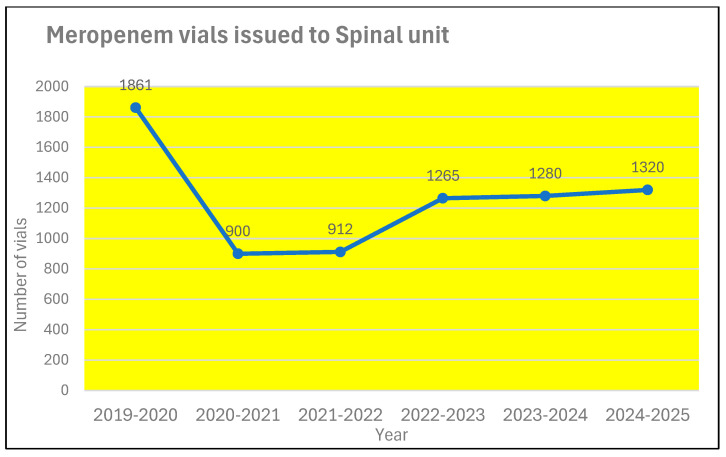
The use of meropenem in the spinal unit during six consecutive twelve-month periods.

**Figure 12 microorganisms-13-02257-f012:**
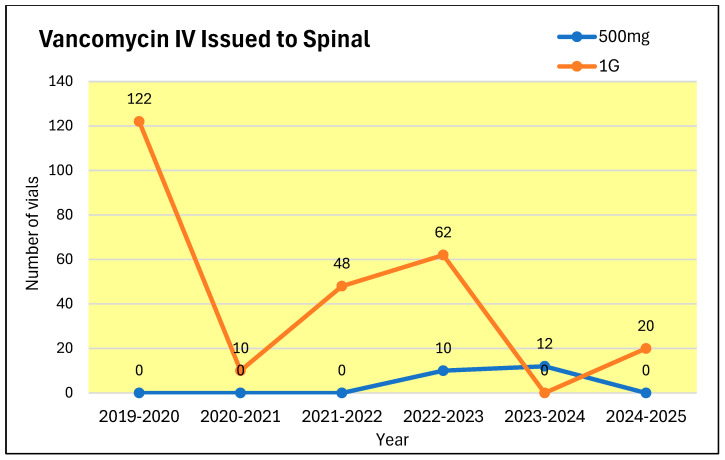
Number of 500 mg and 1 g vials of vancomycin for intravenous use issued to spinal unit patients.

**Figure 13 microorganisms-13-02257-f013:**
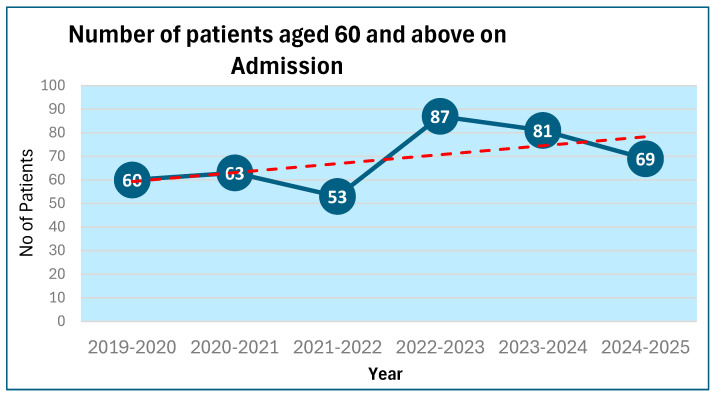
Number of patients aged 60 and above on admission. Red dashed line is the ‘trend line’ showing the trend over the last 6 years, which is an increasing number of patient aged 60 and above on admission.

**Table 1 microorganisms-13-02257-t001:** The antimicrobial resistance totals of the first positive samples per organisms found. The data range is from April 2019 to March 2025. The definition of the abbreviations are R (Resistant), S (Susceptible), I (Intermediate), and N/A (No Data). N/A indicates samples where lab results did not show a result for that antibiotic. This data represents both rectal VRE screening and urine cultures positive for VRE, for both newly colonised patients on the unit and patients with a previous medical history of VRE colonisation from other care units, who have been transferred in.

Antibiotic	Resistance	* **Enterococcus avium** *	* **Enterococcus faecalis** *	* **Enterococcus faecium** *	* **Enterococcus gallinarum** *	***Enterococcus*** sp.	* **Enterococcus** * * **raffinosus** *
Aminoglycoside	R	9	23	146	0	5	11
	S	0	0	0	0	0	0
	I	0	0	0	0	0	0
	N/A	1	6	33	1	1	0
Amoxicillin	R	8	4	173	1	6	10
	S	2	25	2	0	0	0
	I	0	0	0	0	0	0
	N/A	0	0	4	0	0	1
Co-Trimoxazole	R	2	20	84	1	3	5
	S	5	4	32	0	0	3
	I	3	2	49	0	3	2
	N/A	0	3	14	0	0	1
Teicoplanin	R	10	29	179	1	6	10
	S	0	0	0	0	0	1
	I	0	0	0	0	0	0
	N/A	0	0	0	0	0	0
Vancomycin	R	10	29	179	1	6	11
	S	0	0	0	0	0	0
	I	0	0	0	0	0	0
	N/A	0	0	0	0	0	0
Tigecycline	R	0	0	3	0	0	0
	S	10	27	166	1	6	10
	I	0	0	0	0	0	0
	N/A	0	2	10	0	0	1
Linezolid	R	0	0	0	0	0	0
	S	10	28	177	1	6	11
	I	0	0	0	0	0	0
	N/A	0	1	2	0	0	0
Total samples		10	29	179	1	6	11

## Data Availability

The original contributions presented in this study are included in the article. Further inquiries can be directed to the corresponding author.

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
