# Peer review of "Vancomycin-Resistant *Enterococcus* Colonisation in the Patients of a Regional Spinal Cord Injury Unit in Northwest England, United Kingdom: Our Experience with Non-Isolation of VRE Colonised Patients"

_microorganisms, 2025, doi:10.3390/microorganisms13102257_

Round 1
Reviewer 1 Report (Previous Reviewer 1)
Comments and Suggestions for Authors
Dear Authors,
please follow the comments below:
Minor Comments
The manuscript lacks consistent formatting in the figures (in captions of the figures and their X- and Y-axes, etc), therefore, the presentation of the manuscript needs some improvement.
Minor comments are as follows:
- In the title of the manuscript: …E..
- In Figure 4: Apr 2024-Mar 2025. Also, please, enlarge the legends “ Male; Female”. They are now too small.
- L145: remove italics where not needed.
- In Figure 6: Apr 2019-Mar 2025.
- In Figure 8: VRE Colonisation Urine, Apr 2019- Mar 2025 % by Sex. Also, please, enlarge the legends “ Male; Female”. They are now too small.
- In Figure 9: For all year ranges, please, use the following format: 2019-2020, 2020-2021 etc.
- Figures 4 and 8 should provide data for the same annual periods to provide clear information to readers, e.g. Figure 4 requires data on VRE colonisation Rectal from April 2019 to March 2025.
- Please, use the same font size and style for all figure titles. Also, choose and use the same font size and style for the X- and Y-axes of all figures.
A well-prepared manuscript is essential for effective communication of the study. Therefore, minor editorial revisions are necessary before the manuscript can be considered for publication.
Author Response
The manuscript lacks consistent formatting in the figures (in captions of the figures and their X- and Y-axes, etc), therefore, the presentation of the manuscript needs some improvement;
Thank you. Done
- In the title of the manuscript: …E.. : Done
- In Figure 4: Apr 2024-Mar 2025. Also, please, enlarge the legends “ Male; Female”. They are now too small.
- L145: remove italics where not needed.
- In Figure 6: Apr 2019-Mar 2025.
- In Figure 8: VRE Colonisation Urine, Apr 2019- Mar 2025 % by Sex. Also, please, enlarge the legends “ Male; Female”. They are now too small.
- In Figure 9: For all year ranges, please, use the following format: 2019-2020, 2020-2021 etc.
- Figures 4 and 8 should provide data for the same annual periods to provide clear information to readers, e.g. Figure 4 requires data on VRE colonisation Rectal from April 2019 to March 2025.
- Please, use the same font size and style for all figure titles. Also, choose and use the same font size and style for the X- and Y-axes of all figures.
- Thank you. Done
Reviewer 2 Report (Previous Reviewer 4)
Comments and Suggestions for Authors
The abstract is well written. The introduction has been improved. The aim of the paper is clear. The methodology is well described. The results are clearly presented, the charts have been corrected, and the references have been revised.The abstract is well written. The introduction has been improved. The aim of the paper is clear. The methodology is well described. The results are clearly presented, the charts have been corrected, and the references have been revised.
Author Response
The authors thank the reviewer for the valuable comments. Thank you.
Reviewer 3 Report (New Reviewer)
Comments and Suggestions for Authors
Dear editor and authors
Concerning the MS "Vancomycin-resistant enterococcus colonisation in the patients of a regional spinal cord injury unit in northwest England, United Kingdom. Our experience with non-isolation of VRE colonised patients"
Introduction; What is the main knowledge gab of the study, it is not clear?
Methods
- In the method section you need to show how data in figs1-4 were obtained.
- How the VRE were detected among the tested patients.
- What is the AMR pattern of the isolates
- What is the Antimicrobial treatment regimen of the patient?
- What about nurses and doctor, / surfaces colonization.
- Is any of the patients had bowel evacuation in the bed to use this as risk factor lines 305-320
Results/discussion
- Need more organization
- What is the correlation of data added value of figs 1-4
- What is correlation of the VRE colonization and stay length or the level of injury
- The result section requires to add subtitles
- Lines 131-143; is a case study not data for patients , followed by series of cases
- All the VRE are asymptomatic so the risk is low and may be there is other risk factor for inpatients of the spinal unit did not included in the study.
- Very long discussion depends on the previous studied with no data/result evidence from this study.
Minor
The a scientific writing style need to be revised
Using No patient in the MS not represent a scientific writing style
Do not start the sentences with numbers line 278
Author Response
Introduction; What is the main knowledge gab of the study, it is not clear?
Knowledge Gap is stated now.
Methods
- In the method section you need to show how data in figs1-4 were obtained. Stated under Results.
- How the VRE were detected among the tested patients. Stated under Methods.
- What is the AMR pattern of the isolates Please see Table 1 at the end of the manuscript.
- What is the Antimicrobial treatment regimen of the patient? Patients colonised with VRE were not prescribed antibiotics to treat VRE.
- What about nurses and doctor, / surfaces colonization. Nurses and Doctors were not screened for VRE. We did not try to culture VRE from environmental swab.
- Is any of the patients had bowel evacuation in the bed to use this as risk factor lines 305-320. We are studying the role, if any, of bowel evacuation in bed to hospital acquired infections separately.
Results/discussion
- Need more organization
- What is the correlation of data added value of figs 1-4. These figures provide background information on the type of patients we are talking about. For example, many people are not aware that spinal cord injury patients stay in the spinal unit for several weeks.
- What is correlation of the VRE colonization and stay length or the level of injury We are not drawing conclusions as the data may not be adequate to provide a valid conclusion. We are offering the information to readers so that the readers can comprehend the global picture of what happens in the spinal unit. Longer a patient stays in a health care facility, greater the chances of the patient acquiring an infection related to hospitalisation.
- The result section requires to add subtitles Done
- Lines 131-143; is a case study not data for patients , followed by series of cases
- All the VRE are asymptomatic so the risk is low and may be there is other risk factor for inpatients of the spinal unit did not included in the study. We mentioned the risk factors in Introduction: "prolonged hospital stay, presence of long-term indwelling urinary catheters, use of antibiotics for chest, urinary tract infections, the many health care workers with whom they come in contact in a single day "
- Very long discussion depends on the previous studied with no data/result evidence from this study. Wherever relevant, we have stated the observation from this study and compared our findings with those recorded in the literature. For example,
In our spinal unit (where patients colonised with Vancomycin-resistant Enterococcus are not isolated), currently (April 2025), 17 patients (42.5%) are colonised with Vancomycin-resistant Enterococcus in contrast to the 55% patients becoming colonised with Vancomycin-resistant Enterococcus in Cleveland as reported by Dumford [12]."
Minor
The a scientific writing style need to be revised; We have attempted to improve the writing style.
Using No patient in the MS not represent a scientific writing style - ??
Do not start the sentences with numbers line 278. I think I addressed this.
This manuscript is a resubmission of an earlier submission. The following is a list of the peer review reports and author responses from that submission.
Round 1
Reviewer 1 Report
Comments and Suggestions for Authors
Dear Authors,
please find the attached file with comments on the manuscript microorganisms-3658565.

Reviewer 2 Report
Comments and Suggestions for Authors
The research article titled "Vancomycin-resistant enterococcus colonisation in the patients
of a regional spinal cord injury unit in northwest England, United Kingdom. Our experience with non-isolation of VRE colonised patients " have an interesting piece of research, but the presentation of the work must be rigorous for it to be accepted. This work has significant flaws that, in my opinion, prevent it from being published. This is a medical field job, which means you must be rigorous in your writing and work. The results of a study carried out with sick patients are being presented, which implies that their informed consent be included in the methodology. This manuscript has significant information gaps in the introduction and fails to indicate its novelty.
The quality of the figures presented and the discussion of existing medical results should be significantly improved.
the quality of writing and information must be significantly improved.
Reviewer 3 Report
Comments and Suggestions for Authors
Dear Authors,
The Manuscript sent to me contains too many serious errors to be accepted for further publication steps. All parts of it need to be rewritten from scratch. In creating this Article from scratch, the Authors may be guided by the following comments:
Title
In its present form, it is unacceptable. It should be shorter, more digestible and interesting to the reader.
Abstract
It should introduce the topic, i.e., the Authors should include here information about Enterococcus: why it is so dangerous, why it is worth studying, and what risks it poses in the hospital environment, especially this VRE.
This should be followed by a brief description (2-3 sentences) of the Authors' research, results (2-3 sentences) and a concluding sentence at the end.
Abstract in its current form is an abbreviated version of methods and results. There is nothing about the main character of this Manuscript - Enterococcus.
Introduction
It is suitable for rewriting from scratch. It lacks an adequate introduction to the topic studied and the contribution of Enterococcus to it: what infections is it responsible for, what is the current epidemiology? The authors should highlight the validity of their research.
lines 60/61 - if the authors had stated what % was associated with Enterococcus then they could have gently moved on to describing this genus and its importance in nosocomial infections. As it currently presents the topic to the reader, the matter is highly debatable, as Enterococcus is not the most common bacterium transmitted in the hospital setting. So why did the Authors focus specifically on it? The authors should clearly indicate the place that Enterococcus occupies in nosocomial infections. It is worth briefly describing the problem arising from vancomycin resistance. What is its significance in these infections?
In addition, the last paragraph needs to be developed into a general introduction to the research addressed in this Article.
Also: lines 63, 67, 77, 73, 74 - please use italics. Please apply this comment throughout the Manuscript.
In addition, after introducing the abbreviation VRE in line 63, please use only the abbreviation from then on without the full name
lines 74/75 - "Clostridium difficile" - its current correct name is "Clostridioides difficile" - please change
lines 67-69 - please provide a citation for the recommendation
Materials and Methods:
Should include more information about patients: age, gender, other concomitant diseases. They should be divided into sections. Authors should add sections on: Enterococcus culture (medium, incubation conditions), methods (with description) to check VRE resistance, identification methods the authors used to identify per species.
In addition, please provide a citation stating that if Enterococcus is found, a rectal swab is sufficient, not a fecal swab (please cite specific recommendations). Also, please decide whether patients were given rectal swabs (line 83) or stool samples (line 90). There is no consistency. One excludes the other.
Results:
The presentation of these results is completely ill-considered by the authors.
Line 95 - this information should be given in materials and methods.
Fig 1, 2, 3 - please describe the axes of the graph
Fig 2, 3 - I think the designations C1-C4, C5-C8.... should be described in the legend to Fig. They may not be clear to everyone.
line 122 "inpatient"?
lines 131-132 - suddenly specific species appear. Why are they not there before? Why didn't the authors do an analysis of the frequency of isolation of specific Enterococcus species? Without these earlier analyses, this information is redundant and adds nothing.
lines 122 - 160 - should be collected into a general description, without singling out specific patients. This introduces unnecessary confusion.
Fig. 7 - the “Year axis” is unclearly presented. Full years should be given.
'axis title'? In addition, the quality of Fig. is very poor - it should be improved. This also applies to the other Fig.
The results of urinary tract infection - should be put together into one. Previously, the authors state that there is 1 patient in April 2025, and Fig. 7 breaks off at 2024/25 without including this patient. Lack of consistency. Maybe the analysis should be done by the end of 2024 (since 2025 is still ongoing)?
Fig. 10, 11, 12 can be presented in 1 summary of the treatment process.
How do the authors explain the use of vancomycin despite the knowledge that there are patients with isolated VRE strains on the ward?
In addition, for better readability, the authors should introduce sct. from antibiotics.
The authors have a lot of work ahead of them, however, I believe they will improve their Manuscript to a publishable form. I wish them the best of luck.
Kind regards
Reviewer 4 Report
Comments and Suggestions for Authors
Respected Authors,
The topic of the paper is extremely important. I have a few suggestions on how the paper could be further improved. The abstract is well-structured and well-written, as is the introduction. The objective or objectives need to be clearly defined. It is necessary to specify the type of the study that was conducted. Inclusion and exclusion criteria should be provided and added to the paper. Ethical Committee approval is required. The discussion section could include the strengths and limitations of the study. The conclusion is ina greement with the results, but it should be further aligned with the aim of the study.